# Development of the Mobile Technology Vulnerability Scale among Youth and Young Adults Living with HIV

**DOI:** 10.3390/ijerph18084170

**Published:** 2021-04-15

**Authors:** Nadra E. Lisha, Torsten B. Neilands, Xavier A. Erguera, Parya Saberi

**Affiliations:** 1Center for Tobacco Control Research and Education, University of California, San Francisco, CA 94134, USA; nadra.lisha@ucsf.edu; 2Division of Prevention Science, University of California, San Francisco, CA 94158, USA; Torsten.neilands@ucsf.edu (T.B.N.); Xavier.erguera@ucsf.edu (X.A.E.)

**Keywords:** youth living with HIV, technology, mobile telephone, adherence, antiretroviral therapy, scale development

## Abstract

Introduction: Youth and young adults living with HIV (YLWH) in the US have the lowest viral suppression percentage. Lack of sufficient technology access may be correlated with HIV health outcomes in this population. Methods: We developed a Mobile Technology Vulnerability Scale (MTVS; 18 items) among 18–29-year-olds. Exploratory factor analysis (EFA) was performed on baseline data (N = 79), followed by a confirmatory factor analysis (CFA) of 6-month follow-up data (N = 69). Cronbach’s alpha for internal consistency and test-retest reliability were examined. We also correlated the scale with self-report antiretroviral therapy (ART) adherence. Results: EFA yielded a single-factor solution at baseline after dropping one item. CFA at follow-up corroborated the single-factor. Cronbach’s alpha was high and MTVS was correlated with ART adherence at both time points. MTVS at baseline and 6 months were correlated. Conclusion: The 17-item MTVS scale was found to be valid and reliable and related to ART adherence.

## 1. Introduction

Youth and young adults living with HIV (YLWH) in the US have the lowest percentage of viral suppression (as low as 12–58%) [1] compared to older age groups and experience significant health disparities with regard to HIV treatment initiation and clinical outcomes [2,3,4,5,6]. Compared to older adults, YLWH have lower antiretroviral therapy (ART) initiation rates, suboptimal ART adherence and care retention, and higher virologic failure rates [6]. Unsuppressed HIV viral load can result in morbidity, drug resistance, and secondary transmission [7]. Prior studies have shown a significant correlation between the use of technology and virologic suppression and ART adherence among people living with HIV [8,9]. Due to the ubiquity of ownership of technologies (such as smartphones) among youth [10] and the need to address health disparities among YLWH [1,2,3,4,6,11], technology-based methods to facilitate healthcare services and address health disparities are a logical option.

The Mobile Technology Vulnerability Scale (MTVS) was developed based on food insecurity measures. We use a theory applicable to both constructs, the Conservation of Resources Theory [12], which emphasizes that stress can happen when resources are either lost or at risk of being lost. In fact, anticipatory loss can be as bad as actual loss [12]. Accordingly, we believe that as food insecurity is associated with ART non-adherence [13], so is technology vulnerability.

Here, we report steps in the development and validation of a survey instrument designed to examine the level of “mobile technology vulnerability” among YLWH. We believe that a mobile technology vulnerability scale (i.e., a scale to examine the lack of sufficient access to mobile technology to meet personal needs) may be correlated with HIV health outcomes.

## 2. Materials and Methods

Participants and Study Design: We developed the Mobile Technology Vulnerability Scale (MTVS) among 18–29-year-olds living with HIV recruited for a longitudinal pilot trial, evaluating the feasibility and acceptability of a mobile health application called WYZ [14]. MTVS is analogous to the food insecurity scale [15] and examines how secure or stable an individual feels regarding their personal access to mobile technology.

Development: The original mobile technology scale was initially developed by the study Principal Investigator. In an iterative fashion, the scale was reviewed and revised by three psychometrics experts from the UCSF Center for AIDS Prevention Studies (CAPS) Methods Core and one behavioral researcher from the CAPS Developmental Core with expertise in ART adherence and HIV health outcomes. Finally, cognitive interviewing was undertaken with the CAPS Youth Advisory Panel (YAP), whose members are 18–29-year-old YLWH, to further refine the scale by ensuring the cultural appropriateness of the language, enhancing understandability, capturing important issues regarding access and permanency of mobile technology, and overall editing of the scale. Items were revised based on the YAP’s feedback, resulting in an initial 18-item instrument that was deployed in our pilot sample of YLWH.

Measures: At baseline, participants of the pilot study were asked their demographics, including age, sex at birth, sexual orientation, race/ethnicity, financial security (I have enough money to live comfortably, I can barely get by on the money I have, I cannot get by on the money I have), and work status.

The Mobile Technology Vulnerability Scale (MTVS) was developed as an 18-item scale designed to measure how secure an individual feels about their access to mobile technology, namely through their mobile phone (Table 1). Participants responded to MTVS questions at baseline and 6 months.

The self-reported antiretroviral therapy (ART) adherence scale is a composite of three items; “In the last 30 days, on how many days did you miss at least one dose of any of your HIV medications?” (0–30); “In the last 30 days, how good a job did you do at taking your HIV medications in the way you were supposed to?” (0 = very poor, 5 = excellent); and “In the last 30 days, how often did you take your HIV medications in the way you were supposed to?” (0 = rarely to 5 = always) [16]. Participants responded to questions at baseline and 6 months. For analyses, item responses for the three adherence items were linearly transformed and combined to be calculated on a 0–100 scale, with zero being the lowest adherence and 100 the highest [16].

### Statistical Analysis

The sample characteristics were described using means, standard deviations (SD), frequencies, and percentages. To assess the dimensionality of MTVS, we used three complementary techniques: exploratory and confirmatory factor analysis (EFA and CFA, respectively) in M*plus* 8.2 [17] and Cronbach’s alpha. An EFA analysis was carried out on the baseline data with a GEOMIN (oblique) rotation of the factor structure to achieve a simpler structure with greater interpretability. A weighted least square estimation approach appropriate for the binary items (M*plus* estimator WLSMV) was used to extract the factor(s). To determine the number of factors to retain, we considered the following criteria: a factor loading cutoff criteria of 0.32 or larger [18], scree plots of Eigenvalues, model fit chi-square statistics, and three commonly used fit statistics from the structural equation modeling (SEM) literature: root mean square error of the approximation (RMSEA), comparative fit index (CFI), and standardized root mean squared residual (SRMR). We considered any two of the following criteria indicating satisfactory fit: RMSEA ≤ 0.06, CFI ≥ 0.95, SRMR ≤ 0.08, or a non-significant chi-square (i.e., *p* ≥ 0.05) [19].

Importantly, we considered the interpretability of each EFA solution. We examined solutions with between 1 and 9 factors. Internal consistency was examined by computing Cronbach’s alpha. We also examined validity using the Pearson correlation of the scale with self-reported ART adherence. We examined the stability of the factor structure using a confirmatory factor analysis (CFA) with data at the 6-month follow-up. Lastly, because our expectation was that technology vulnerability would be a relatively stable construct in our population during this 6-month interval, we explored test-retest reliability via a Pearson correlation of the two total scale scores from baseline and 6 months.

## 3. Results

Demographics: The sample consisted of 79 study participants with a mean age of 27.0 (SD = 2.9), who were predominantly male (87%), Latino (43%), non-Hispanic Black (21%), and non-heterosexual (89%). The sample mostly worked (36% full time, 22% part-time), and 67% stated that they could “barely get by” or could “not get by” on the money that they had. At baseline and 6 months, ART adherence was 85% at both time points.

EFA and CFA: For the EFA of baseline data, we determined that the best fitting model is a single-factor solution as this factor was clearly dominant based on strong loadings on all the items except one (Table 1). Item three loaded below our cutoff of 0.32 (loading = 0.25); therefore, we dropped this item and re-ran the EFA with the remaining items. Without item three, the results also indicated a single-factor solution. The Eigenvalues, examined via a scree plot, showed a clear drop off after the first factor. In addition, the one-factor model had a non-significant chi-square (χ^2^ = 133.3, df = 119, *p* = 0.17), indicating good model fit and all the items loaded above 0.35, with most being above 0.50. RMSEA (0.04, 90% confidence interval (CI) = 0.00, 0.07; CFI = 0.98; and SRMR = 0.19) also satisfied the SEM criteria for good fit. Lastly, the CFA using a single factor model with the 6-month data, indicated good fit via RMSEA (0.04, 90% CI = 0.00, 0.07; CFI = 0.99; SRMR = 0.16; and χ^2^ = 131.3, df = 119, *p* = 0.21).

Cronbach’s alpha, validity, and test-retest reliability: Next, we examined the internal consistency reliability of the scale using Cronbach’s alpha at baseline and 6 months. MTVS exhibited a strong alpha of 0.84 (baseline) and 0.90 (six months). We created a MTVS score by taking the mean of the 17 items and correlated that score with self-reported ART adherence. MTVS was correlated with the self-reported ART adherence (baseline: r = −0.31, *p* = 0.008, six months: r = −0.55, *p* < 0.0001). In other words, there was a statistically significant correlation between increased technology vulnerability and decreased ART adherence. The Pearson’s correlation between MTVS at baseline and 6 months was 0.51 (*p* < 0.0001), supporting our expectation that technology vulnerability was stable in our population across the 6-month measurement period of the study.

## 4. Discussion

To our knowledge, this is the first report of a mobile technology vulnerability scale and its association with ART adherence. YLWH are disproportionately affected by health disparities and often exhibit lower adherence to ART. As part of a pilot trial examining the usefulness of a mobile health application targeting these health disparities, we examined a new scale, MTVS, as a possible correlate of HIV health outcomes.

The results indicate that the data best fit a single-factor solution, both using an EFA at baseline and a CFA at 6 months. The CFA revealed high factor loadings with only a few items having smaller loadings (i.e., items 1 and 14). The internal consistency of the scale was excellent, as evaluated using Cronbach’s alpha. The scale appeared valid as it exhibited a negative correlation with self-reported ART adherence in the expected direction, indicating that more technology vulnerability was related to lower ART adherence. The scale also demonstrated good test-retest reliability as baseline MTVS was correlated with 6-month MTVS.

Limitations of study include a convenience sample and relatively small sample size of YLWH living and/or receiving health services in the San Francisco Bay Area, which limits the generalizability of our findings. Additionally, we relied on self-reported ART adherence, which is subject to recall and social desirability biases and may represent an over-estimation of actual ART adherence. However, overestimation of adherence by self-report measures would lower variance in adherence, which would weaken its correlation with MTVS, suggesting that MTVS could be even more strongly correlated with biomarker-based measures of adherence. Future studies should evaluate this possibility using biomarkers of ART adherence.

As YLWH have lower rates of ART initiation and adherence resulting in unsuppressed HIV viral load, it is important to identify which factors are associated with these health outcomes. MTVS appears to be a valid and reliable scale that is associated with ART adherence. Future research should examine further refinements of MTVS questions with moderate loadings, the association of MTVS with other objective HIV health outcomes, the use of MTVS in other populations, and the impact of interventions aimed to increase MTVS on improved ART adherence. Exploring other aspects of construct validity (e.g., convergent/divergent validity) should also be studied in future research.

Among youth and young adults, technology is nearly ubiquitous [10]. In addition, younger age groups have lower ART adherence versus older age groups [6], thus, it is possible that the association between MTVS and ART adherence is more prominent in the younger age groups.

## 5. Conclusions

Factor analyses of MTVS yielded a single factor. The resulting scale is reliable and related to ART adherence. MTVS might be useful in future research to identify subgroups of the population who may experience vulnerability regarding their personal access to technology and subsequent downstream health outcomes, such as low medication adherence. In clinical practice, medical professionals may consider asking patients about their mobile technology vulnerability and referring them to programs that can assist in acquiring the needed technology.

## Figures and Tables

**Table 1 ijerph-18-04170-t001:** Items in the Mobile Technology Vulnerability Scale (MTVS), frequencies, and exploratory factor analysis (EFA) and confirmatory factor analysis (CFA) results.

	n (%)	EFA Rotated Factor Loadings (All Items) at Baseline	EFA Rotated Factor Loadings (without Item 3) at Baseline	CFA Standardized Factor Loadings at 6 Months
1. In the last 6 months, I was the only person who used this phone (not including lending to someone to make a brief phone call or to look something up on the internet).	70 (91%)	0.41	0.41	0.46
2. At any time in the last 6 months, I received formal assistance to pay for my cellphone service (such as Lifeline Assistance Program/Obama Phone).	9 (12%)	0.57	0.57	0.74
3. At any time in the last 6 months, my family or friends helped me to pay for my cell phone service.	32 (42%)	0.26	-	-
4. At any time in the last 6 months, I had more than one cell phone number.	10 (13%)	0.53	0.53	0.81
5. At any time in the last 6 months, my cell phone was stolen at least once.	10 (13%)	0.61	0.60	0.70
6. At any time in the last 6 months, I lost my cell phone at least once.	12 (15%)	0.61	0.61	0.68
7. At any time in the last 6 months, my cell phone service was disconnected (cut off) at least once because I didn’t pay the bill.	17 (22%)	0.78	0.78	0.85
8. At any time in the last 6 months, I did not pay other bills (example: utilities, rent, etc.) so I could pay my cell phone bill.	10 (13%)	0.89	0.89	0.92
9. At any time in the last 6 months, I did not buy necessary items (example: food, clothes, medication, etc.) so I could pay my cell phone bill.	14 (18%)	0.83	0.84	0.90
10. At any time in the last 6 months, I did not pay my cell phone bill because I had to pay for other necessities or other bills.	15 (20%)	0.73	0.73	0.91
11. At any time in the last 6 months, I had to limit using my cell phone’s data plan for any purpose (such as making calls, sending text messages, or using the internet) so that I could keep my cell phone bill low.	13 (17%)	0.67	0.67	0.80
12. At any time in the last 6 months, I used free internet services (such as Google Voice, WhatsApp, or Facebook Messenger’s phone option) to make phone calls because I did not have cell phone service.	24 (32%)	0.97	0.97	0.96
13. At any time in the last 6 months, I checked email, sent text messages, checked social media, searched the internet, or made a call on my cell phone by using free public Wi-Fi because I could not afford to use my data plan.	24 (32%)	0.88	0.88	0.95
14. At any time in the last 6 months, I had to use a less reliable (example: Boost, Cricket, etc.) cell phone service because it was cheaper than other more reliable services.	13 (17%)	0.35	0.35	0.90
15. At any time in the last 6 months, I did not make an important phone call because I was frustrated with my phone’s service.	7 (9%)	0.78	0.78	0.92
16. At any time in the last 6 months, I did not search for important information that I needed because I was frustrated with my phone’s internet connection.	16 (21%)	0.80	0.79	0.90
17. Over the last 6 months, I had personal problems (such as missed an appointment, got lost, was unable to pay a bill, etc.) because my cell phone battery died.	17 (23%)	0.82	0.82	0.93
18. Over the last 6 months, I had problems (such as missed an appointment, got lost, was unable to pay a bill, etc.) because I didn’t pay my cell phone bill and my cellphone service was cut.	7 (10%)	0.88	0.88	0.90

Notes: Confirmatory factor analysis (CFA) (N = 69); exploratory factor analysis (EFA) (N = 79). Missing value numbers ranged from 2 to 8 people. Item 1 was reverse coded. Item 3 was removed from the final scale. Factor loadings were estimated via a weighted least squares (WLS) approach in M*plus* (M*plus* WLSMV estimator). CFA global model fit test result: χ^2^(119) = 131.298, *p* = 0.21.

## Data Availability

The dataset generated and analyzed during the current study are available from the corresponding author on reasonable request.

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
