# Peer review of "Development of the Mobile Technology Vulnerability Scale among Youth and Young Adults Living with HIV"

_ijerph, 2021, doi:10.3390/ijerph18084170_

Round 1

Reviewer 1 Report

Evaluation:

The manuscript presents the development of the Mobile Technology Vulnerability Scale for HIV youth and youth adults. The authors showed EFA followed CFA in 6-month follow-up data. The concept is quite novel, and the results are critical enough to support the author's hypothesis.

Major comments:

1. The concept presented in the manuscript is novel. How did you exclude people living with HIV with a substance use disorder tend to receive inadequate medical care (special in youth or youth adults)?

2. The hypothesis is designed for MTVS in 18 items. Even EFA dropped one item had only 17 items for calculation. The data is enough to demonstrate the authors conclusion?

3. Suggestion to use a three phase approach to refine and evaluate the MTVS. A time period at 1-, 3-, and 6-months should be comprehensive?

4. As the authors stated, unsuppressed HIV viral load can result in morbidity, drug resistance, and secondary transmission. HIV biomarkers can support this hypothesis.

Reviewer 2 Report

This is a preliminary validation study involving measures of 'mobile technology vulnerability' and its relationship with self-reported adherence to HIV medications among individuals living with HIV. The sample size is adequate for the analyses undertaken (n=79 with 6-month follow-up data available for 69 individuals), and significant associations were noted between HIV medication adherence and access to mobile technology.     

The methodology is sound and the approach to validating the MTVS tool is also appropriate, utilising an existing model for assessing food insecurity. In this respect, this study provides an interesting and innovative approach to understanding and potentially improving HIV medication adherence.

The use of language is lucid throughout, and the manuscript is generally well prepared.

In terms of specific comments, further demographic information in Table form or within the text would be valuable. While this study is designed to address the needs of youth living with HIV (ie ages 18-29 years), the population sample appears skewed towards older individuals with an average age of 27 years (standard deviation 2.9). In this respect, the application of this tool to larger clinical populations may be limited by demographic factors such as age and ethnicity that are not able to be addressed adequately here. Are the authors able to comment on the impact of age on these analyses, and particularly the association between MTVS score and medication adherence?        
